# Norsesquiterpenes from the Latex of *Euphorbia dentata* and Their Chemical Defense Mechanisms against *Helicoverpa armigera*

**DOI:** 10.3390/molecules28237681

**Published:** 2023-11-21

**Authors:** Tong An, Dongxu Cao, Yangyang Zhang, Xiamei Han, Zhiguo Yu, Zhixiang Liu

**Affiliations:** 1College of Plant Protection, Shenyang Agricultural University, Shenyang 110866, China; antong@syau.edu.cn; 2College of Biological Science and Technology, Shenyang Agricultural University, Shenyang 110866, China; cdx@syau.edu.cn (D.C.); 2021170128@stu.syau.edu.cn (Y.Z.); a2021170060@stu.syau.edu.cn (X.H.)

**Keywords:** chemical defense, *Euphorbia dentata*, isolate, latex, norsesquiterpenes

## Abstract

*Euphorbia dentata* (Euphorbiaceae), an invasive weed, is rarely eaten by herbivorous insects and could secrete a large amount of white latex, causing a serious threat to local natural vegetation, agricultural production and human health. In order to prevent this plant from causing more negative effects on humans, it is necessary to understand and utilize the chemical relationships between the latex of *E. dentata* and herbivorous insects. In this study, three new norsesquiterpenes (**1**–**3**), together with seven known analogues (**4**–**10**), were isolated and identified from the latex of *E. dentata.* All norsesquiterpenes (**1**–**10**) showed antifeedant and growth-inhibitory effects on *H. armigera* with varying levels, especially compounds **1** and **2**. In addition, the action mechanisms of active compounds (**1**–**3**) were revealed by detoxifying enzyme (AchE, CarE, GST and MFO) activities and corresponding molecular docking analyses. Our findings provide a new idea for the development and utilization of the latex of *E. dentata*, as well as a potential application of norsesquiterpenes in botanical insecticides.

## 1. Introduction

Plants and herbivorous insects are important components of terrestrial ecosystems. Herbivorous insects need to adapt and depend on the host plants for survival, and these host plants affect the feeding of herbivorous insects through their chemicals. The two form many complex adaptive mechanisms in the process of long-term co-evolution [1]. Plant latex, usually stored in laticifers, is an opaque viscous liquid that can be secreted immediately when plants are subjected to herbivorous feeding insects or mechanical damage [2,3]. Plant latex contains a large number of secondary metabolites with diverse types and complex structures. Most of the metabolites have certain antifeedant and growth-inhibitory effects on herbivorous insects, acting as a chemical defense in resisting the feeding of herbivorous insects [4,5]. However, the secondary metabolites of most plant latexes against herbivorous insects have not been systematically studied, and their defense mechanisms against herbivorous insects are less researched. More research findings are needed to support it.

*Euphorbia* is the largest genus in the Euphorbiaceae family, with more than 2000 species worldwide. It is mainly distributed in tropical and subtropical regions of Africa and America [6]. In addition, *Euphorbia* is the largest genus of plants with latex. Some of the latexes of this genus have been used to kill fish or large animals [7]. The main characteristic of *Euphorbia* is that their latexes are rich in secondary metabolites, especially terpenoids with complex structural frameworks, including monoterpenes, sesquiterpenes, diterpenes, and triterpenes [6]. Among them, sesquiterpenes are naturally defensive compounds in plants, which have significant antifeedant and growth-inhibitory effects on herbivorous insects [8].

*Euphorbia dentata* Michx. is an annual plant of the genus *Euphorbia* in the family Euphorbia. It originated in North America and has spread to many countries in Europe, Australia, and Asia [9]. In recent years, the plant has grown rapidly in China and has a strong adaptability to new habitats. Some single-dominant communities have formed in Changsha of Hunan Province, Shijiazhuang and Baoding of Hebei Province, and Guanling of Guizhou Province, causing great harm to local natural vegetation, agricultural production, and human health. Field observations found that the whole plant of *E. dentata* contained white latex, which will flow out of the plant, especially when the plant is damaged by herbivorous insects. Therefore, we speculated that the strong invasive ability of the plant may be closely related to the successful chemical defense mechanisms of its latex against herbivorous insects. However, studies on secondary metabolites of *E. dentata* have not been conducted. Since *E. dentata* belongs to the genus *Euphorbia*, we hypothesized that *E. dentata* is also rich in sesquiterpenes.

*Helicoverpa armigera* (Lepidoptera: Noctuidae) is a herbivorous insect widely distributed in China and other areas of the world. The species is a euryphagous insect, and its host plants have more than thirty families [10]. In recent years, with the adjustment of agricultural planting structures, the occurrence and harm of *H. armigera* have become increasingly serious, causing great economic losses to local agriculture [11]. To effectively control *H. armigera*, extensive research has been carried out in chemical pesticides. Although some progress has been made, it has not stopped the large-scale trend of worldwide harm that *H. armigera* is causing. In addition, due to the long-term dependence on chemical pesticides, *H. armigera* has developed a strong resistance, and the ecological environment damage caused by chemical pesticides has become increasingly serious [12]. Therefore, it is vital to search for safe and effective defensive substances from chemical interactions between plant and herbivorous insects.

In the process of adapting to external stress, insects could produce a class of detoxifying enzymes that could metabolize a large number of foreign substances. Acetylcholinesterase (AchE), carboxylesterase (CarE), glutathione-*S*-transferase (GST), and mixed-function-oxidase (MFO) are important detoxifying enzymes in insects, playing an important role in metabolizing toxic compounds and maintaining normal physiological activities [13,14]. In addition, molecular docking analysis has become an important technique for exploring the interactions between molecules, providing visual analyses to explore the binding sites and modes of action of enzymes and compounds [15]. Therefore, it is necessary to study the defense mechanisms of compounds by detoxifying enzyme activities and corresponding molecular docking analyses.

Field investigations found that *H. armigera* could feed on a small amount of *E. dentata*. At the same time, *E. dentata* secretes white latex from its damaged area. In addition, the latex of *E. dentata* and the methanol extract of latex have significant antifeedant effects on *H. armigera* (Appendix A), which also proved that chemical defense is the main form of defense against *H. armigera.* These phenomena suggest that the latex of *E. dentata* has a chemical defense mechanism against *H. armigera.* However, the secondary metabolites from the latex of *E. dentata* and their defense mechanisms against *H. armigera* are still unknown. In this study, we isolated and identified norsesquiterpenes (Figure 1) from the latex of *E. dentata*. Additionally, the antifeedant and growth-inhibitory effects of norsesquiterpenes on *H. armigera* were also investigated. In addition, the action mechanisms of active norsesquiterpenes were revealed by detoxifying enzyme activities and corresponding molecular docking analyses.

## 2. Results

### 2.1. Structural Elucidation

Norsescycldione A (**1**) was obtained as a yellowish oil with C_13_H_14_O_6_ (seven degrees of unsaturation) using its HR-ESIMS data (*m*/*z* 289.0628 [M + Na]^+^, calculated for C_13_H_14_O_6_Na, 289.0688). Its 1D-NMR spectra (Table 1) showed four carbonyl groups [*δ*_C_ 210.5 (C-2), 200.4 (C-9), 192.3 (C-5), 171.3 (C-11)], two trans-double bond groups [*δ*_H_ 7.82 (1H, s, H-4), 7.24 (1H, d, *J* = 15.8 Hz, H-7), 6.37 (1H, d, *J* = 15.8 Hz, H-8); *δ*_C_ 160.8 (C-4), 147.6 (C-7), 142.1 (C-3), 132.5 (C-8)], one tertiary alcohol group [*δ*_C_ 97.2 (C-6)], and three methyl groups [*δ*_H_ 2.34 (3H, s, H-10), 1.17 (3H, s, H-12), 0.96 (3H, s, H-13); *δ*_C_ 27.2 (C-10), 25.3 (C-12), 23.1 (C-13)]. In the HMBC spectrum (Figure 2), the key correlations between H-4/C-2, C-5, C-11; H-7/C-1, C-5, C-9; H-10/C-8, C-9; and H-12/C-2, C-13 established its planar structure. Compared with the literature, it was found that compound **1** possessed an ionone-type norsesquiterpene skeleton [16]. In addition, the computed ECD curve of (6*S*)-**1a** matched well with the experimental result of **1** (Figure 3), establishing its stereochemical structure. Thus, the structure of **1** was confirmed and named Norsescycldione A (Figure 1).

Norsescycldione B (**2**) was obtained as a yellowish oil with C_13_H_14_O_5_ (seven degrees of unsaturation) using its HR-ESIMS data (*m*/*z* 273.0736 [M + Na]^+^, calculated for C_13_H_14_O_5_Na, 273.0739). Its 1D-NMR spectra (Table 1) showed four carbonyl groups [*δ*_C_ 212.3 (C-2), 204.5 (C-9), 194.0 (C-5), 172.6 (C-11)], two trans-double bond groups [*δ*_H_ 7.75 (1H, s, H-4), 7.28 (1H, dd, *J* = 15.3 Hz, H-7), 6.28 (1H, d, *J* = 15.3 Hz, H-8); *δ*_C_ 161.4 (C-4), 144.2 (C-7), 140.5 (C-3), 136.0 (C-8)], and three methyl groups [*δ*_H_ 2.37 (3H, s, H-10), 1.19 (3H, s, H-12), 0.98 (3H, s, H-13); *δ*_C_ 28.2 (C-12), 27.3 (C-10), 26.1 (C-13)]. A comparison of its 1D NMR data with Norsescycldione A indicated that they were very similar except for one less hydroxyl group, which showed that compound **2** also possessed a norsesquiterpene skeleton similar to Norsescycldione A. In the HMBC spectrum (Figure 2), the key correlations of H-4/C-2, C-5, C-11; H-7/C-1, C-5, C-9; H-10/C-8, C-9; H-12/C-2, C-13 established its planar structure, which also proved that the missing hydroxyl group was at the position of C-6. In addition, the computed ECD curve of (6*R*)-**2a** matched well with the experimental result of **2** (Figure 3), establishing its stereochemical structure. Thus, the structure of **2** was confirmed and named Norsescycldione B (Figure 1).

Norsescycldione C (**3**) was obtained as a yellowish oil with C_13_H_16_O_4_ (six degrees of unsaturation) using its HR-ESIMS data (*m*/*z* 259.0949 [M + Na]^+^, calculated for C_13_H_16_O_4_Na, 259.0946). Its 1D-NMR spectra (Table 1) showed three carbonyl groups [*δ*_C_ 207.1 (C-2), 202.6 (C-9), 195.6 (C-5)], two trans-double bond groups [*δ*_H_ 7.22 (1H, d, *J* = 16.1 Hz, H-7), 6.77 (1H, s, H-4), 6.41 (1H, d, *J* = 16.1 Hz, H-8); *δ*_C_ 146.2 (C-7), 144.2 (C-3), 136.0 (C-4), 134.1 (C-8)], one tertiary alcohol group [*δ*_C_ 99.3 (C-6)], and four methyl groups [*δ*_H_ 2.42 (3H, s, H-10), 2.26 (3H, s, H-11), 1.18 (3H, s, H-12), 0.95 (3H, s, H-13); *δ*_C_ 27.5 (C-10), 25.7 (C-12), 23.9 (C-13), 14.7 (C-11)]. A comparison of its 1D NMR data with Norsescycldione A indicated that they were very similar except for one less carboxyl group and one more methyl group, showing that compound **2** also possessed a norsesquiterpene skeleton similar to Norsescycldione A. In the HMBC spectrum (Figure 2), the key correlations of H-4/C-2, C-5, C-11; H-7/C-1, C-5, C-9; H-10/C-8, C-9; H-12/C-2, C-13; H-11/C-2 established its planar structure, which also proved that the missing carboxyl group was at the position of C-3 and the extra methyl group was attached to the position of C-3. In addition, the computed ECD curve of (6*S*)-**3a** matched well with the experimental result of **3** (Figure 3), establishing its stereochemical structure. Thus, the structure of **3** was confirmed and named Norsescycldione C (Figure 1).

The other compounds **4**–**10** were known and identified as (3*S*, 5*R*, 6*S*, 7*E*)-3,5,6-Trihydroxy-7-megastigmen-9-one (**4**) [17], Elaeocarpunone (**5**) [18], Dehydrovomifoliol (**6**) [19], 3-Hydroxy-5*α*, 6*α*-epoxy-*β*-ionone (**7**) [20], Grasshopper ketone (**8**) [21], (6*R*, 7*E*, 9*R*)-9-Hydroxy-4,7-megastigmadien-3-one (**9**) [17] and 4,5-Dihydroblumenol A (**10**) [22], respectively, based on the NMR data and literature comparison.

### 2.2. Antifeedant and Growth-Inhibitory Effects

It is well known that sesquiterpenes are naturally defensive compounds in plants and have significant antifeedant and growth-inhibitory effects on herbivorous insects [8]. From this, we speculated that the norsesquiterpenes (**1**–**10**) may be the potential defense substances for the latex of *E. dentata* against herbivorous insects. Field investigations found that *H. armigera* (a model herbivorous insect for evaluating chemical defense function) [23,24] could feed on a small amount of *E. dentata*. At the same time, *E. dentata* secretes white latex from its damaged area. These phenomena suggest that the latex of *E. dentata* has a defense mechanism against *H. armigera.* However, the secondary metabolites from the latex of *E. dentata* and their defense mechanisms against *H. armigera* are still unknown. Therefore, we studied their chemical defense functions (antifeedant and growth). As shown in Table 2, norsesquiterpenes (**1**–**10**) had antifeedant and growth-inhibitory effects with varying levels (100, 50, 25 μg/mL). In general, compounds **1**–**3** exhibited a stronger chemical defense function (antifeedant and growth-inhibitory effects) compared with other compounds (**4**–**10**). Among these, compounds **1** and **2** exhibited significant antifeedant effects at 100 μg/mL (85.16 ± 7.44% and 80.62 ± 6.55%, respectively), which was even comparable to neem oil (92.28 ± 7.11%). Moreover, compounds **1** and **2** also showed potent growth-inhibitory effects at 100 μg/mL (74.28 ± 8.35% and 78.11 ± 6.26%, respectively), and interestingly, they did not cause the death of *H. armigera* compared with the positive control.

### 2.3. Detoxifying Enzymes Effects

In the process of adapting to external stress, insects could produce a class of detoxifying enzymes that could metabolize a large number of foreign substances. AchE, CarE, GST, and MFO are important detoxifying enzymes in insects, playing an important role in metabolizing toxic compounds and maintaining normal physiological activities [13,14]. Therefore, the action mechanisms of active norsesquiterpenes (**1**–**3**) were revealed by detoxifying enzyme activities. The effects of compounds **1**–**3** on AchE, CarE, GST, and MFO activities of *H. armigera* at different times (0, 6, 12, 24, 48 h) are shown in Figure 4. In general, after treatment with different compounds (**1**–**3**), the AchE and GST activities of *H. armigera* decreased with the extension of time, MFO activities of *H. armigera* increased with the extension of time, and CarE activities of *H. armigera* increased first and then decreased with the extension of time. Among these, the changes in AchE, CarE, and MFO activities in each treatment group were not significantly different from those in the blank control. It is worth noting that GST activities significantly decreased by 35.54%, 18.41%, 41.25%, and 56.79% under compound **1** treatment; 13.99%, 35.60%, 54.16%, and 44.15% under compound **2** treatment; and 15.49%, 33.63%, 33.12%, and 31.42% under compound **3** treatment compared with the blank controls at 0–6, 6–12, 12–24 and 24–48 h, respectively.

### 2.4. Molecular Docking Analyses

In recent years, molecular docking analysis has become an important technique for exploring interactions between molecules (such as enzymes and compounds) [15]. The binding energies of compounds **1** and **2** with GST of *H. armigera* (−75.8056 and −82.0594 kcal/mol) were lower than those of compound **3** (−60.6084 kcal/mol) (Appendix A). As shown in Figure 5, carbonyl groups of compounds **1**–**3** interacted with neighboring amino acids via hydrogen bonds, indicating that the carbonyl groups of norsesquiterpenes could play an important role in GST activities. In addition, compound **1** formed four hydrogen bonds with amino acids ARG68 (distance 2.73 Å), GLU66 (distance 1.96 Å), SER67 (distance 2.07 Å), and VAL54 (distance 2.17 Å) in GST. Compound **2** formed five hydrogen bonds with amino acids ARG68 (distance 2.50 Å), GLU66 (distance 2.39 Å), PPO55 (distance 2.51 Å), SER67 (distance 2.08 Å), and TRP65 (distance 2.01 Å) in GST. Compound **3** formed three hydrogen bonds with amino acids ARG68 (distance 2.64 Å), GLU66 (distance 1.93 Å), and SER67 (distance 2.64 Å) in GST.

## 3. Discussion

Recently, researchers have paid more attention to the chemical relationships between invasive plants and native herbivorous insects [25,26]. This will help researchers to better reveal the invasion mechanism of alien plants from the perspective of chemical defense, and provides a new idea for the development and utilization of invasive plants. In our study, three new norsesquiterpenes (**1**–**3**), together with seven known analogues (**4**–**10**), were isolated from the latex of *E. dentata* and their structures were determined via HR-ESIMS verifications, NMR analyses, and ECD calculations. Among these, compounds **4**, **8**, and **9** were obtained from the *Euphorbia* genus for the first time. Moreover, this was also the first systematic study of the chemical composition of this plant. All norsesquiterpenes (**1**–**10**) showed chemical defense functions in *H. armigera* with varying levels (100, 50, 25 μg/mL), especially compounds **1** and **2**. These results indicate that the norsesquiterpenes (**1**–**3**) from the latex of *E. dentata* could function as chemical defense substances against *H. armigera*, which may help *E. dentata* to gain a competitive advantage over other plants, as well as supporting theory for the defensive functions of sesquiterpenes against herbivorous insects [8]. By analyzing the structure–activity relationships of norsesquiterpenes (**1**–**10**), the presence of one carboxyl group at the position of C-3 may have a positive effect on chemical defense function. However, more studies of norsesquiterpene analogues are needed to confirm this.

In addition, in the detoxifying enzyme activities, the GST activities significantly decreased under treatment with compounds **1**–**3** compared with a blank control with an extension of time. This indicated that GST may be a key target of *H. armigera* for compounds **1**–**3** to exert their chemical defense functions. GST is an important detoxifying enzyme in insects, which could catalyze the binding of harmful substances and reduce glutathione, thereby increasing the water solubility of harmful substances and excreting them [13]. Norsesquiterpenes (**1**–**3**) can significantly inhibit the GST activity of *H. armigera*, which indicates that compounds **1**–**3** may block the excretion of harmful substances, resulting in the inhibition of the feeding and growth of *H. armigera*. On this basis, we explored the binding sites and modes of action of GST, as well as compounds **1**–**3,** using molecular docking analyses. The binding energies of compounds **1** and **2** with GST of *H. armigera* were lower than those of compound **3**, indicating that compounds **1** and **2** bound to GST more stably than compound **3**. This was essentially consistent with the above results (antifeedant and growth-inhibitory effects of compounds **1**–**3**). It is well-known that hydrogen bonds play a more important role than other forces in interactions between molecules [27]. Compounds **1** and **2** formed more hydrogen bonds with amino acids than compound **3**, which not only verified that more hydrogen bonds led to a more stable binding between enzyme and compound, but also indicated that the common amino acids (ARG68, GLU66, and SER67) could be key active sites of GST interacting with compounds **1**–**3**.

## 4. Materials and Methods

### 4.1. General

Column chromatography was carried out using silica gel (Qingdao Marine, Qingdao, China) and Sephadex LH-20 (GE Healthcare, Chicago, IL, USA). Preparative HPLC was carried out using a 1220 system (Agilent, Santa Clara, CA, USA) equipped with a 5C18-MS-II column (COSMOSIL, Tokyo, Japan). UV spectra were carried out using a 241 spectrophotometer (Perkin Elmer, Waltham, MA, USA). HR-ESIMS spectra were carried out using a 6545 Q-TOF instrument (Agilent, USA). NMR spectra were carried out using an AV-600 instrument (Bruker, Saarbrucken, Germany). ECD spectra were carried out using a MOS-450 instrument (Bio-Logic, Seyssinet Pariset, France).

### 4.2. Plant Material

*E. dentata* was identified by Professor Bo Qu and its voucher specimen (NO. ZW-2020-0073) and was kept in Shenyang Agricultural University. The latex of *E. dentata* (aerial part) was collected from Beijing Botanical Garden, China (40°01′ E, 116°21′ N) in August 2020.

### 4.3. Insect Material

*H. armigera* was purchased from Henan Keyun Bio-Pesticide Co., Ltd. (Henan, China) and identified by Associate Professor Lu Jiang (Shenyang Agricultural University).

### 4.4. Extraction and Isolation

The latex of *E. dentata* (2 L) was suspended with 70% methanol (2 L) under an ultrasonic bath for 20 min and then centrifuged at 12,000 rpm for 10 min. The supernatant was merged and concentrated in vacuo. The concentrated extract (11 g) was subjected to a silica gel column (normal-phase, dichloromethane/methanol, 98:2–80:20, *v*/*v*) to give seven subfractions (Fr. A–Fr. F). Fr. B (110 mg) was subjected to a preparative HPLC (210 nm, methanol/H_2_O, 25:75, *v*/*v*, 5.0 mL/min) to give compounds **6** (9.1 mg, *t*_R_ = 38.2 min) and **9** (13.5 mg, *t*_R_ = 30.1 min), respectively. Fr. C (460 mg) was subjected to a Sephadex LH-20 column (methanol/H_2_O, 80:20, *v*/*v*) to give five subfractions (Fr. C-1–Fr. C-5). Fr. C-2 (61 mg) was subjected to a preparative HPLC (210 nm, methanol/H_2_O, 10:90, *v*/*v*, 5.0 mL/min) to give compounds **4** (13.4 mg, *t*_R_ = 53.7 min). Fr. C-3 (125 mg) was subjected to a preparative HPLC (210 nm, acetonitrile/H_2_O, 10:90, *v*/*v*, 5.0 mL/min) to give compounds **5** (2.5 mg, *t*_R_ = 77.3 min), **7** (9.5 mg, *t*_R_ = 58.1 min), and **10** (6.2 mg, *t*_R_ = 67.4 min), respectively. Fr. D (430 mg) was subjected to a Sephadex LH-20 column (methanol/H_2_O, 90:10, *v*/*v*) to give four subfractions (Fr. D-1–Fr. D-4). Fr. D-2 (83 mg) was subjected to a preparative HPLC (210 nm, acetonitrile/H_2_O, 10:90, *v*/*v*, 5.0 mL/min) to give compounds **2** (5.8 mg, *t*_R_ = 37.5 min) and **8** (1.4 mg, *t*_R_ = 48.0 min), respectively. Fr. E (246 mg) was subjected to a preparative HPLC (210 nm, methanol/H_2_O, 10:90, *v*/*v*, 5.0 mL/min) to give compounds **1** (7.5 mg, *t*_R_ = 33.6 min) and **3** (8.4 mg, *t*_R_ = 44.1 min), respectively.

### 4.5. Spectroscopic Data

Norsescycldione A (**1**): yellowish oil; ECD (methanol) *λ*_max_ (Δ*ε*) 205 (+21.40), 215 (+20.10), 239 (−82.47), 254 (+52.85), 324 (+9.42) nm; HR-ESIMS at *m*/*z* 289.0628 [M + Na]^+^ (calculated for C_13_H_14_O_6_Na, 289.0688); ^1^H and ^13^C NMR data; see Table 1.

Norsescycldione B (**2**): yellowish oil; ECD (methanol) *λ*_max_ (Δ*ε*) 201 (+31.26), 215 (+8.76), 242 (−34.99), 259 (+60.66), 310 (+12.12) nm; HR-ESIMS at *m*/*z* 273.0736 [M + Na]^+^ (calculated for C_13_H_14_O_5_Na, 273.0739); ^1^H and ^13^C NMR data; see Table 1.

Norsescycldione C (**3**): yellowish oil; ECD (methanol) *λ*_max_ (Δ*ε*) 207 (+48.86), 241 (−105.20), 259 (+83.52), 302 (+5.96) nm; HR-ESIMS at *m*/*z* 259.0949 [M + Na]^+^ (calculated for C_13_H_16_O_4_Na, 259.0946); ^1^H and ^13^C NMR data; see Table 1.

(3*S*, 5*R*, 6*S*, 7*E*)-3,5,6-Trihydroxy-7-megastigmen-9-one (**4**): yellowish oil; ^1^H and ^13^C NMR data, see Appendix A.

Elaeocarpunone (**5**): yellowish oil; ^1^H and ^13^C NMR data; see Appendix A.

Dehydrovomifoliol (**6**): yellowish oil; ^1^H and ^13^C NMR data; see Appendix A.

3-Hydroxy-5*α*, 6*α*-epoxy-*β*-ionone (**7**): yellowish oil; ^1^H and ^13^C NMR data; see Appendix A.

Grasshopper ketone (**8**): yellowish oil; ^1^H and ^13^C NMR data, see Appendix A.

(6*R*, 7*E*, 9*R*)-9-Hydroxy-4,7-megastigmadien-3-one (**9**): yellowish oil; ^1^H and ^13^C NMR data; see Appendix A.

4,5-Dihydroblumenol A (**10**): yellowish oil; ^1^H and ^13^C NMR data; see Appendix A.

### 4.6. ECD Calculations

The ECD calculations were carried out by a previously reported method [25]. The conformational analyses of compounds **1**–**3** were performed using Spartan 14.0 software under the MMFF94 force field. The obtained conformations were optimized using Gaussian 09 software at the B3LYP/6-31G (d) level. Theoretical calculations were performed using TDDFT at the B3LYP/6-311+G (2d, p) level in methanol. The final ECD spectra were generated using SpecDis 1.60 software based on the Boltzmann weighting of each conformation.

### 4.7. Antifeedant and Growth-Inhibitory Assay

The antifeedant assay was carried out according to a previously reported method [26]. The leaf discs (1 cm in diameter from the leaves of *Brassica chinensis*) of the treatment group were painted with latex (50 µL) of *E. dentata*, methanol extract of latex (50 µL, 0.5 mg/mL), or methanol solution (20 µL) containing different concentrations of compounds (100, 50, and 25 μg/mL). The blank control group was painted with the same amount of methanol. The positive control group was painted with the same amount of commercial insecticide (neem oil). After natural drying, four discs (two treated and two blank controls) were placed in a Petri dish at a crosswise position. Two third-instar *H. armigera* (starved for 6 h) were placed in each Petri dish. When about 80% of the leaf discs (blank control) were consumed, the *H. armigera* was removed from the Petri dish, and the consumed area of the leaf discs was measured. Each assay was repeated at least five times. Antifeedant rate (%) was calculated as (A_C_ − A_T_)/Ac × 100 (A_C_, the consumed area of leaf discs in the blank control group; A_T_, the consumed area of leaf discs in the treatment group).

The growth-inhibitory assay was carried out according to a previously reported method [28]. The leaf discs (1 cm in diameter from the leaves of *B. chinensis*) of the treatment group were painted with methanol solution (20 µL) containing 100 μg/mL compounds. The blank control group was painted with the same amount of methanol. The positive control group was painted with the same amount of commercial insecticide (neem oil). After natural drying, ten discs (ten treated or ten controls) were placed in a Petri dish. A weighed third-instar *H. armigera* (starved for 6 h) was placed in each Petri dish. The *H. armigera* were accurately weighed at 24, 48, and 72 h, respectively. Each assay was repeated at least five times. The growth-inhibitory rate (%) was calculated as (W_C_ − W_T_)/Wc × 100 (W_C_, the gained weight of *H. armigera* in the blank control group; W_T_, the gained weight of *H. armigera* in the treatment group).

The leaf discs were pretreated as the growth-inhibitory assay above. After natural drying, ten discs (ten treated or ten controls) were placed in a Petri dish. A third-instar *H. armigera* (starved for 6 h) was placed in each Petri dish. The activities of detoxifying enzymes were measured at 0, 6, 12, 24, and 48 h, respectively. The extraction of enzymes was carried out according to a previously reported method [29]. Third-instar *H. armigera* with similar sizes were weighed, and nine volumes of normal saline were added. The obtained homogenate was then centrifuged at 12,000 rpm for 10 min, and the supernatant was used for the follow-up enzyme solution.

### 4.8. Detoxifying Enzymes Assay

The AchE assay was carried out according to Ellman’s method [30]. The enzyme solution (100 μL) was mixed with acetylthiocholine iodide (0.075 M, 100 μL). The mixture was incubated at 30 °C for 15 min and terminated with 5, 5′-dithiobis-(2-nitrobenzoic acid) (0.01 M, 50 μL). The absorbance was detected at 405 nm. The CarE assay was carried out according to Asperen’s method [31]. Enzyme solution (100 μL) was mixed with *α*-naphthyl acetate (0.03 M, 100 μL). The mixture was incubated at 30 °C for 15 min and terminated with diazo blue lauryl sulphate (0.01 M, 50 μL). The absorbance was detected at 600 nm. The GST assay was carried out according to Habig’s method [32]. The enzyme solution (100 μL) was mixed with 1-chloro-2, 4-dinitrobenzene (0.001 M, 100 μL) and glutathione (0.001 M, 100 μL). The absorbance was detected at 340 nm. The MFO assay was carried out according to Shang’s method [33]. The enzyme solution (100 μL) was mixed with paranitroanisole (0.05 M, 100 μL). The mixture was incubated at 30 °C for 15 min and terminated with HCl (0.01 M, 50 μL). The absorbance was detected at 400 nm. Each enzyme assay was repeated at least five times. One unit of enzyme activity was expressed as a change of 0.001 absorbance per milligram of protein per minute.

### 4.9. Molecular Docking Analyses

The molecular docking analyses were carried out according to a previously reported method [26]. The 3D structure of the enzyme (GST of *H. armigera*) was constructed via homology modeling using the EasyModeller 4.0 software. The GST sequence of *H. armigera* was obtained from the NCBI database in FASTA format (GenBank NO. BK40535.1). The optimal template (PDB code: 3VK9) was searched using the BLAST server based on E value, sequence identity, and query coverage (Appendix A). The 3D structures of compounds **1**–**3** were optimized by the ChemDraw-3D 14.0 software to minimize energy. The Molegro Virtual Docker 4.0 software was carried out for further docking analyses. An active pocket (radius = 15.0) centered on x (39.90), y (33.15), and z (39.55) was formed based on the binding sites of the original ligand (glycerol) and enzyme (3VK9).

## 5. Conclusions

In conclusion, three new norsesquiterpenes (**1**–**3**), together with seven known analogues (**4**–**10**), were isolated from the latex of *E. dentata* using multiple column chromatography and preparative HPLC. Additionally, their structures were identified using HR-ESIMS verifications, NMR analyses, and ECD calculations. All norsesquiterpenes (**1**–**10**) showed chemical defense functions on *H. armigera* with varying levels (100, 50, 25 μg/mL), especially compounds **1** and **2** (antifeedant effects: 85.16 ± 7.44% and 80.62 ± 6.55% at 100 μg/mL, growth-inhibitory effects: 74.28 ± 8.35% and 78.11 ± 6.26% at 100 μg/mL, respectively). In addition, the action mechanisms of active compounds (**1**–**3**) were revealed via detoxifying enzymes (AchE, CarE, GST and MFO) activities and corresponding molecular docking analyses. GST activities significantly decreased under treatment with compounds **1**–**3** compared with the blank control with the extension of time. The common amino acids (ARG68, GLU66, and SER67) could be key active sites of GST interacting with compounds **1**–**3**. These results will help researchers to better reveal the secret effects of latex from *E. dentata* against herbivorous insects from the perspective of chemical defense, providing new ideas for the development and utilization of the latex of *E. dentata.*

## Figures and Tables

**Figure 1 molecules-28-07681-f001:**
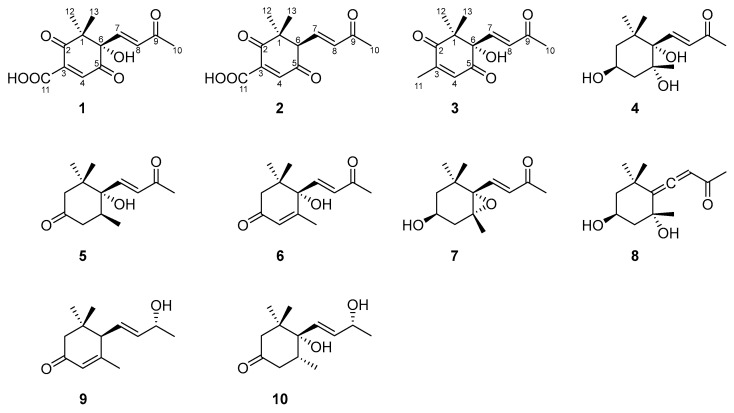
Structures of compounds **1**–**10**.

**Figure 2 molecules-28-07681-f002:**
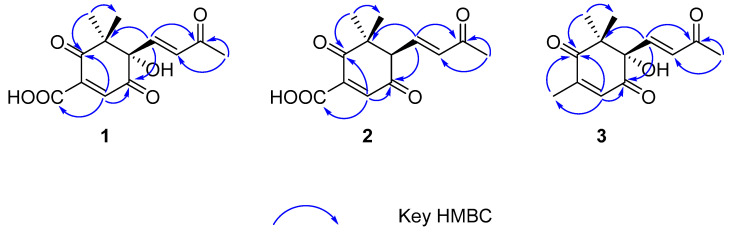
Key HMBC correlations of compounds **1**–**3**.

**Figure 3 molecules-28-07681-f003:**
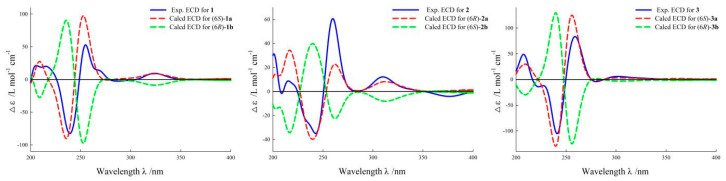
ECD curves of compound **1**–**3**.

**Figure 4 molecules-28-07681-f004:**
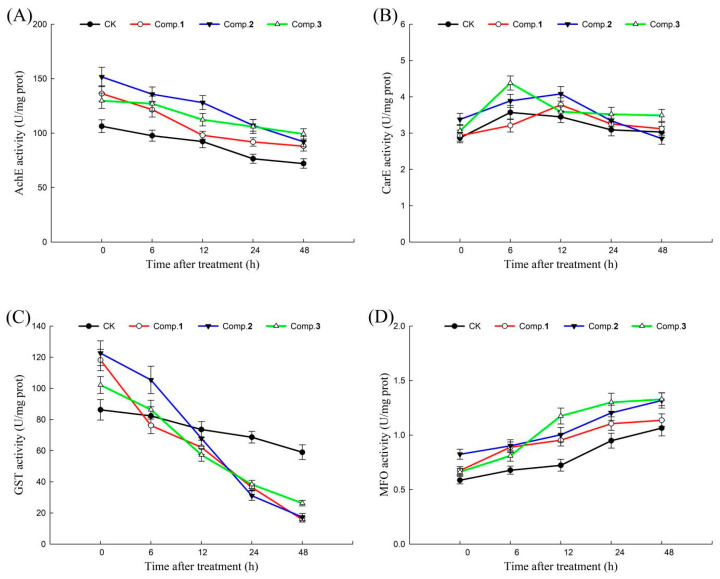
The effects of compounds **1**–**3** (100 μg/mL) on the detoxifying enzymes of *H. armigera*: (**A**) the effects of compounds **1**–**3** on AchE; (**B**) the effects of compounds **1**–**3** on CarE; (**C**) the effects of compounds **1**–**3** on MFO; and (**D**) the effects of compounds **1**–**3** on GST. CK = Control check.

**Figure 5 molecules-28-07681-f005:**
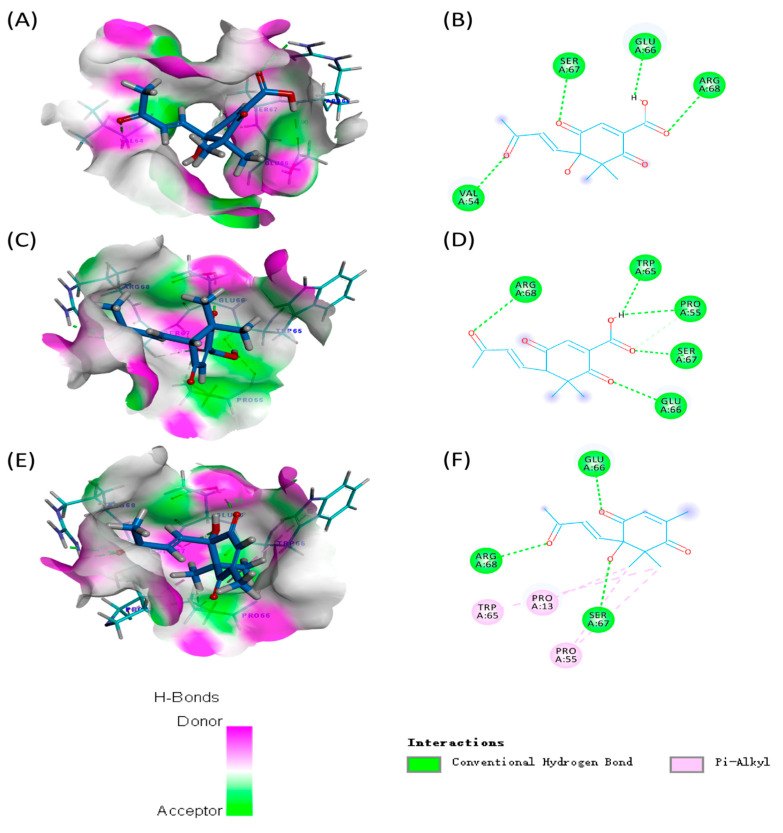
Molecular docking analyses of compounds **1**–**3** with GST: (**A**) 3D diagram of the interactions between compound **1** and the active pocket of GST; (**B**) 2D diagram of the interactions between compound **1** and amino acids of GST; (**C**) 3D diagram of the interactions between compound **2** and the active pocket of GST; (**D**) 2D diagram of the interactions between compound **2** and amino acids of GST; (**E**) 3D diagram of the interactions between compound **3** and the active pocket of GST; and (**F**) 2D diagram of the interactions between compound **3** and amino acids of GST.

**Table 1 molecules-28-07681-t001:** 1H (600 MHz) and 13C NMR (150 MHz) data of compounds **1**–**3** in methanol-*d*4.

Position	1		2		3	
	*δ* _C_	*δ* _H_	*δ* _C_	*δ* _H_	*δ* _C_	*δ* _H_
1	51.7		44.9		52.4	
2	210.5		212.3		207.1	
3	142.1		140.5		144.2	
4	160.8	7.82 (1H, s)	161.4	7.75 (1H, s)	136.0	6.77 (1H, s)
5	192.3		194.0		195.6	
6	97.2		55.8	2.96 (1H, d, 8.9)	99.3	
7	147.6	7.24 (1H, d, 15.8)	144.2	7.28 (1H, dd, 15.3, 8.9)	146.2	7.22 (1H, d, 16.1)
8	132.5	6.37 (1H, d, 15.8)	136.0	6.28 (1H, d, 15.3)	134.1	6.41 (1H, d, 16.1)
9	200.4		204.5		202.6	
10	27.2	2.34 (3H, s)	27.3	2.37 (3H, s)	27.5	2.42 (3H, s)
11	171.3		172.6		14.7	2.26 (3H, s)
12	25.3	1.17 (3H, s)	28.2	1.19 (3H, s)	25.7	1.18 (3H, s)
13	23.1	0.96 (3H, s)	26.1	0.98 (3H, s)	23.9	0.95 (3H, s)

**Table 2 molecules-28-07681-t002:** Antifeedant and growth-inhibitory effects of compounds **1**–**10** on *H. armigera*.

Comp.	Antifeedant Rate (%)			Growth-Inhibitory Rate (%)		
	100 µg/mL	50 µg/mL	25 µg/mL	100 µg/mL (24 h)	100 µg/mL (48 h)	100 µg/mL (72 h)
**1**	85.16 ± 7.44	68.14 ± 7.82	42.19 ± 3.67	33.18 ± 2.73	50.31 ± 4.79	74.28 ± 8.35
**2**	80.62 ± 6.55	73.20 ± 7.11	40.17 ± 4.69	24.37 ± 2.81	44.15 ± 3.93	78.11 ± 6.26
**3**	57.14 ± 6.28	42.63 ± 5.02	35.89 ± 4.22	45.16 ± 6.02	53.19 ± 5.71	60.43 ± 5.79
**4**	33.15 ± 4.90	28.12 ± 3.16	12.18 ± 1.73	6.32 ± 0.89	17.25 ± 1.94	29.15 ± 3.43
**5**	19.51 ± 2.16	13.70 ± 1.12	10.04 ± 0.87	ND ^b^	ND ^b^	ND ^b^
**6**	17.27 ± 1.19	15.46 ± 1.45	9.17 ± 1.02	4.15 ± 0.26	13.04 ± 1.56	15.24 ± 2.01
**7**	14.55 ± 2.01	10.57 ± 1.82	4.26 ± 0.53	3.57 ± 0.43	7.29 ± 0.81	11.65 ± 1.32
**8**	ND ^b^	ND ^b^	ND ^b^	ND ^b^	ND ^b^	ND ^b^
**9**	23.35 ± 1.96	14.18 ± 2.03	8.77 ± 0.85	4.75 ± 0.68	10.12 ± 1.35	24.21 ± 1.94
**10**	34.25 ± 2.98	13.90 ± 1.17	5.96 ± 0.64	6.12 ± 0.83	13.48 ± 1.57	21.80 ± 1.95
Neem oil	92.28 ± 7.11	86.45 ± 6.93	62.17 ± 3.85	ND ^a^	ND ^a^	ND ^a^

ND = No detected (^a^: *H. armigera* were dead within 24 h, ^b^: the content of compound was not enough).

## Data Availability

All details and data can be found in the text.

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
