# Peer review of "Norsesquiterpenes from the Latex of Euphorbia dentata and Their Chemical Defense Mechanisms against Helicoverpa armigera"

_molecules, 2023, doi:10.3390/molecules28237681_

Round 1

Reviewer 1 Report

Comments and Suggestions for Authors

The authors present a manuscript entitled “Norsesquiterpenes from the latex of Euphorbia dentata and their chemical defense mechanisms against Helicoverpa armigera”. The plus of this manuscript is discovery of dinorsesquiterpenes from the latex of Euphorbia dentata which are relatively rare. The structure determination of new compounds was performed on the basis of spectroscopy studies including NMR, MS but without specific optical rotation. The ECD (both experimental and calculations) used to determine their absolute configuration are plus. The biological activity study of the compounds was performed using antifeedant and growth-inhibitory assays. The assay was designed to assess the function of the compounds as chemical defense. The result was that the new compounds have stronger activities reported as in percentage.  No statistical parameter e.g., EC50, ED50 were reported. Other advantages are that the authors performed molecular docking analysis and investigation of detoxifying enzyme effects. In conclusion, the manuscript is interesting and can be accepted after revision. The specific optical rotation should be completed. It is a plus if the authors can express the result of assays using statistical parameter. The natural products isolated from Euphorbia dentata should be enriched to give the status of the current discovery. Introduction can be improved. Finally, many grammatical errors are found in manuscript suggesting the English check is required.

Comments on the Quality of English Language

many grammatical errors are found in manuscript suggesting the English check is required.

Author Response

The specific optical rotation should be completed.

Thank you very much for your good suggestions. However, after testing the activities, there are no enough compounds to measure optical rotation. For the structure confirmation of compounds, we tested HR-ESIMS, 1D-NMR, 2D-NMR and ECD calculation, which are more accurate and reproducible than optical rotation.

It is a plus if the authors can express the result of assays using statistical parameter.

Thank you very much for your good suggestions. The EC50 and ED50 values are calculated by drawing a concentration-activity curve. At least five concentrations are required to calculate the EC50 and ED50 values. However, we only set three concentrations in the activity, which is not enough to accurately calculate the EC50 and ED50 values. And there are no enough compounds to measure activities again.

The natural products isolated from Euphorbia dentata should be enriched to give the status of the current discovery.

Thank you very much for your good suggestions. We have enriched the natural products isolated from Euphorbia dentata. “However, the studies on secondary metabolites of E. dentata have not been reported. Since E. dentata belongs to the genus Euphorbia, we hypothesized that E. dentata is also rich in sesquiterpenes.”

Introduction can be improved.

Thank you very much for your good suggestions. We have improved the introduction.

Finally, many grammatical errors are found in manuscript suggesting the English check is required.

Thank you very much for your good suggestions. We have followed the suggestions and carefully modified the grammatical errors in the manuscript.

Reviewer 2 Report

Comments and Suggestions for Authors

In the article titled "Norsesquiterpenes from the latex of Euphorbia dentata and their chemical defense mechanisms against Helicoverpa armigera," the authors investigate Euphorbia dentata (Euphorbiaceae), an invasive plant rarely consumed by herbivorous insects. In this study, they isolate and identify three new norsesquiterpenes and seven known analogues from the latex of E. dentata. All the norsesquiterpenes exhibited antifeedant and growth inhibitory effects on H. armigera at various levels. The authors assessed the activity of some of these compounds through the action of detoxifying enzymes (AchE, CarE, GST, and MFO) and conducted a molecular docking analysis.

In my opinion, I recommend the manuscript for publication in Molecules, but there are some recommendations for the authors:

In the introduction, the authors should specify what the main detoxifying enzymes are and explain their mechanisms of action. They should also clarify why they chose to experimentally analyze these detoxifying enzymes.

In Figure 4, please indicate the meaning of "CK."

In the extraction and isolation section, the authors should specify whether the silica used was of direct or reverse phase. It is recommended that the authors create a diagram illustrating the fractionation of E. dentata latex and include it as a figure to enhance understanding of the experimental process.

The authors should engage in a more in-depth discussion of their results. There is a notable lack of cited bibliographical references.

Regarding the molecular docking analysis, the authors mentioned that the structures were drawn using ChemDraw-3D 14.0 software, but they did not specify whether these structures were optimized to accurately represent the three-dimensional structure of the compounds. How can the authors ensure that the structures provided by the software are suitable for the analysis conducted?"

Author Response

In the introduction, the authors should specify what the main detoxifying enzymes are and explain their mechanisms of action. They should also clarify why they chose to experimentally analyze these detoxifying enzymes.

Thank you very much for your good suggestions. We have specifed what the main detoxifying enzymes are and explained their mechanisms of action. “In the process of adapting to external stress, insects could produce a class of detoxifying enzymes that could metabolize a large number of foreign substances.  Acetylcholinesterase (AchE), Carboxylesterase (CarE), Glutathione-S-transferase (GST) and Mixed-function-oxidase (MFO) are important detoxifying enzymes in insects, which play an important role in metabolizing toxic compounds and maintaining normal physiological activities [13, 14].  In addition, molecular docking analysis has become an important technique for exploring the interactions between molecules, providing visual analyses to explore the binding sites and modes of action of enzymes and compounds [15]. Therefore, it is necessary to study the defense mechanisms of compounds by detoxifying enzymes activities and corresponding molecular docking analyses.”

In Figure 4, please indicate the meaning of "CK."

Thank you very much for your good suggestions. We have indicated the meaning of "CK." “CK = Control check.”

In the extraction and isolation section, the authors should specify whether the silica used was of direct or reverse phase.

Thank you very much for your good suggestions. We have specified the silica used was normal-phase. “The concentrated extract (11 g) was subjected to a silica gel column (normal-phase, dichloromethane/methanol, 98: 2–80: 20, v/v) to give seven subfractions (Fr. A–Fr. F). “

It is recommended that the authors create a diagram illustrating the fractionation of E. dentata latex and include it as a figure to enhance understanding of the experimental process.

Thank you very much for your good suggestions. We have added the flowchart of extraction and isolation in the supporting information.

The authors should engage in a more in-depth discussion of their results. There is a notable lack of cited bibliographical references.

Thank you very much for your good suggestions. We have engaged in a more in-depth discussion of results. “These results indicated that the norsesquiterpenes (1−3) from the latex of E. dentata could function as chemical defense substances against H. armigera, which may help E. dentata to gain a competitive advantage over other plants, as well as supporting theory for the defensive functions of sesquiterpenes against herbivorous insects [8].” “GST is an important detoxifying enzyme in insects, which could catalyze the binding of harmful substances and reduced glutathione, thereby increasing the water solubility of harmful substances and making them excreted [13].  Norsesquiterpenes (1−3) could significantly inhibit the GST activity of H. armigera, which indicated that compounds 1−3 may block the excretion of harmful substances, resulting in the inhibition of feeding and growth of H. armigera.”

Regarding the molecular docking analysis, the authors mentioned that the structures were drawn using ChemDraw-3D 14.0 software, but they did not specify whether these structures were optimized to accurately represent the three-dimensional structure of the compounds. How can the authors ensure that the structures provided by the software are suitable for the analysis conducted?"

Thank you very much for your good suggestions. We have specified these structures optimized by ChemDraw-3D 14.0 software. “The 3D structures of compounds 1−3 were optimized by the ChemDraw-3D 14.0 software to minimize energy.”

Round 2

Reviewer 1 Report

Comments and Suggestions for Authors

I can understand if no optical rotation because there are ECD exp and calculation although it is highly recommended to provide specific optical rotation for dereplication work.  For known compounds, it is more accurate to compare not only NMR data but also their specific optical rotations. Anyhow, the current data is acceptable if the authors are not able to provide more. 

Author Response

Thank you very much for your good advice and understanding. In future experiments, we should first measure optical rotation and then measure activity, so as to better ensure the accuracy of data.

Reviewer 2 Report

Comments and Suggestions for Authors

1. The authors say they have included the latex fractionation scheme in the supporting information (I did not find it mentioned in the manuscript) 

"Supplementary Materials: Supplementary materials are available online. Figure S1−S29: NMR spectrum of compounds 110. Figure S30: The superimposed 3D structures of GST target and template. Figure S31: Antifeedant effects of the latex of E. dentata on H. armigera. Figure S32: Antifeedant effects of the methanol extract of latex on H. armigera. Table S1: The optimal template for 3D structure modelling of GST. Table S2: Moldock scores of the compounds (13) with GST of H. armigera."

I suggest that it be placed within the manuscript.

2. There should be a more in-depth discussion of the results.

Author Response

(1) Thank you very much for your good advice. We have added "Figure S33: The flowchart of extraction and isolation of the latex of E. dentata." in the manuscript.

(2)Thank you very much for your good advice.  We have added a in-depth discussion in the manuscript.

Round 3

Reviewer 2 Report

Comments and Suggestions for Authors

The authors have improved the manuscript.

As a whole I consider the manuscript worthy to be published